# Peste des petits ruminants virus virulence is associated with an early inflammatory profile in the tonsils and cell cycle arrest in lymphoid tissue

Roger-Junior Eloiflin,[1,2,3] Llorenç Grau-Roma,[4] Vincent Lasserre,[1,2] Sylvie Python,[5,6] Stephanie Talker,[5,6] Philippe Totte,[1,2] Obdulio García-Nicolás,[5,6] Artur Summerfield,[5,6] Arnaud Bataille[1,2]

**ABSTRACT**  Using a systems immunology approach, this study comprehensively explored the immunopathogenesis of peste des petits ruminants (PPR) focussing on strain-dependent differences in virulence. Saanen goats were infected either with the highly virulent (Morocco 2008 [MA08]) or the low-virulent (Ivory Coast 1989 [IC89]) strain of the PPR virus (PPRV). As expected, MA08-infected goats exhibited higher clinical scores, pronounced lymphocyte depletion, and lesions affecting mucosal and lymphoid tissues. CD4 T cells were more affected in terms of depletion and infection in peripheral blood. Transcriptional analyses of the blood and lymphoid tissue demonstrated activation of interferon type I (IFN-I) responses at 3 days post-infection (dpi) only with MA08, but comparable IFN-I expression levels with MA08 and IC89 at 6 dpi. MA08 strain induced strong inflammatory and myeloid cell-related transcriptional responses observed in tonsils but not in mesenteric lymph node. This inflammatory response in the tonsils was associated with an extensive damage and infection of the tonsillar epithelium in the crypts, pointing to a barrier defect as a possible cause of inflammation. An early and prominent downregulation of cell cycle gene networks was observed in all compartments analyzed in MA08-infected animals. This effect can be interpreted as suppressed lymphocyte proliferation that may cause immunosuppression during the first week following MA08 infection. A proteome analysis confirmed synthesis of IFN-I response proteins during infection with both strains, but only MA08 strain additionally upregulated ribosomal and inflammation-related proteins. In conclusion, the present comprehensive investigation delineates strain-dependent differences in early immunopathological processes associated with severe inflammation disease and a blunted lymphocyte proliferation.

**IMPORTANCE** Field observations show that the severity of PPR is highly dependent on the viral (PPRV) strains and the host infected, but the mechanisms behind these variations are not well understood. Here we compare immune response in Saanen goats infected with high (MA08) and low (IC89) virulent PPRV strains. Analyses revealed a differential immune response: early activation of IFN-I responses only with MA08 but comparable IFN-I expression levels with MA08 and IC89 at later stages. Additionally, MA08 strain triggered inflammatory and myeloid cell-related responses in the tonsils and marked suppression of lymphocyte proliferation evidenced by cell cycle arrest. CD4 T cells were found to be most affected in terms of depletion in the peripheral blood. Massive infection of the tonsils seems to induce epithelial lesions that promote the inflammatory responses. These results underscore the need to understand strain-specific differences for PPR surveillance and control.

Address correspondence to Arnaud Bataille, arnaud.bataille@cirad.fr.

Artur Summerfield and Arnaud Bataille contributed equally to this article.

The authors declare no conflict of interest.

See the funding table on p. 15.

**KEYWORDS** peste des petits ruminants, virulence, immunopathogenesis, goat, morbillivirus

In the realm of infectious diseases affecting domestic small ruminants and wild artiodactyls, peste des petits ruminants (PPR) remains an ubiquitous and devastating threat, presenting a significant concern for the global economy, food security, and biodiversity (1–3). The causative pathogen is the PPR virus (PPRV), a member of the *Morbillivirus* genus, capable of inducing acute and subacute clinical manifestations of the disease. PPRV is mainly transmitted through direct contact between healthy and infected animals or by contaminated aerosols. Indirect transmission may also occur through contaminated feed, water, and fomites, although the importance of such transmission is not yet well understood (4).

The susceptibility of host species to PPRV infection is characterized by marked heterogeneity, with symptoms observed and disease outcome dependent on the virus strain and the host species and breed infected (5). This heterogeneity is of significant epidemiological relevance and intimately tied to the complex interplay between the host organism and the specific viral strains involved (6, 7), but it is unclear how this is associated with differences in innate and adaptive immune responses. Availability of validated diagnostic tests, as well as cheap and efficient vaccines, is essential to Food and Agriculture Organization (FAO) and World Organisation for Animal Health (WOAH) efforts to eradicate PPR by 2030 (8, 9). Nevertheless, efficiency of surveillance and control strategies are also dependent on improving our understanding of host susceptibility and pathogenesis.

The receptor for PPRV is the signaling lymphocyte activation molecule, as with other morbilliviruses (10), which is mainly expressed by activated lymphocytes. PPRV infection of lymphocytes may prevent clonal expansion of virus-specific lymphocytes, thereby suppressing the development of adaptive immune responses, as demonstrated for other morbilliviruses (11–16). On the other hand, several innate immune mechanisms, such as autophagy (17), inflammasome (18), induction of the interferon type I (IFN-I) response (19), inhibition of nucleotide biosynthesis (20), and apoptosis (21), have been shown *in vitro* to help limit the replication of the virus. Nevertheless, innate mechanisms, particularly virus-induced inflammation, may also play a role in strain-dependent differences in virulence (22).

To understand these differences in PPRV virulence, we have established a model using two virulent PPRV strains and Saanen goats. The Ivory Coast 1989 (IC89) PPRV strain belongs to PPRV lineage I, which is mainly found in West Africa. The IC89 induces per acute disease in the West African Djallonké breeds of goat and sheep (6, 22, 23). The Morocco 2008 (MA08) PPRV strain belongs to lineage IV, which has historically been found in Asia and has spread in Middle East and North Africa. The MA08 strain was highly pathogenic in alpine goats (24, 25). In Saanen goats, the MA08 PPRV induced severe disease, lymphocyte depletion, prominent macroscopic and histological lesions, high viremia, and high virus shedding during the acute phase of disease, whereas the IC89 strain did not (26). A first *in vitro* experiment had already shown that differential immune response could be observed in peripheral blood mononuclear cells (PBMCs) in this challenge model (22). Considering the importance of the immune response in morbillivirus pathogenesis, the present study aimed to characterize *in vivo* the differences in the induction of innate immune responses and the initiation of adaptive immune responses in the peripheral blood and lymphoid tissues between MA08 and IC89. To this end, we used blood samples collected during a first experiment (7), and we performed a second experiment using the same model to collect secondary lymphoid tissues. These are expected to be the primary target of PPRV at early time points post-infection in order to understand the relationship between innate immune responses and pathogenesis. To achieve this aim, we employed a multiomics approach including flow cytometry, RNA sequencing, and mass spectrometry. Our data demonstrate that virulence is associated

with a dysregulated inflammatory response and prominent suppression of cell proliferation in lymphoid tissue.

## RESULTS

### Viral RNA and nucleoprotein in organs and cells

Quantification of viral RNA reads in RNA sequencing data of PBMCs obtained from the first goat experiment (experiment layout depicted in Fig. 1A) revealed the presence of viral gene expression already at 3 days post-infection (dpi) in animals infected with strain MA08, with clearly increased levels at 7 dpi and decreasing levels at 12 dpi (Fig. 2A). In contrast, samples from IC89-infected animals had lower viral loads that were only detected at 7 and 12 dpi (Fig. 2A). Different organs and cells collected during the second animal experiment were investigated in a similar manner for the presence of viral RNA (experiment layout depicted in Fig. 1B). This confirmed the presence of viral RNA in sampled organs from MA08-infected animals already at 3 dpi with a further increase at 6 dpi (Fig. 2B). In contrast, in IC89-infected animals, viral RNA was detected in only one organ of one animal at 3 dpi (Fig. 2B). In all compartments, MA08-infected animals had the highest number of viral reads. Our data also indicate a stronger viral presence in the tonsils than in the superior mesenteric lymph node (MLN) for both viral strains (Fig. 2B; see also confirmatory reverse transcription quantitative PCR for viral RNA in Fig. S1).

Lymphocyte subpopulation dynamics monitored by flow cytometry revealed a significant depletion of CD4 T cells (CD6$^+$CD4$^+$) starting at 5 dpi with MAO8. This was not observed following infection with the IC89 strain. The CD8 T cells showed only mild depletion at the end of the observation period, which appeared to be similar with both PPRV strains. CD6-MHCII cells were not affected (Fig. 2C). PPRV nucleoprotein expression was predominantly found in the CD4 T cells starting at 5 dpi, specifically during MA08 infection (Fig. 2D).

### PPRV strain-dependent innate immunity transcriptional profiling

#### PBMCs and white blood cells

PBMCs and white blood cells (WBCs) that were collected in the two experiments showed similar trends on the modulation of innate immunity with a clear discrimination between the two viral strains (Fig. 3A and B). At 3 dpi, the initiation of an IFN-I response was found only in the goats infected with the MA08 strain (e.g., blood transcriptional modules [BTMs] M150 in PBMCs and M68 and M111.0 in WBCs; Fig. 3A). Nevertheless, this response appeared to be more prominent at the later time points (6 and 7 dpi) with the IC89 strain, which induced more IFN-I BTM such as M111.1, M75, and M127. At 12 dpi, the IFN-I BTM returned to steady state with both groups (Fig. 3A).

Analyses of the BTM related to inflammation and myeloid cells revealed a major difference in the host response between the two PPRV strains (Fig. 3A). Inflammatory and myeloid cell-related BTMs (M33, M53, M37.0, M11.1, M87, M25, S4, M118.0, and M11.0) showed activation in D7/D3 (in PBMCs) and D6/D3 (in WBCs) comparisons in MA08-infected animals only (Fig. 3A and B, respectively). Interestingly, the D7/D0 and D6/D0 did not show such a strong induction in these BTMs (Fig. 3A and B, respectively). To explain this apparent contradiction, we had a look at the normalized enrichment scores (ESs) for inflammatory and myeloid cell BTM at 3 dpi and found that most of them were negative but did not reach statistical significance. IC89 only induced platelet activation (M32.1) at 7 dpi in the PBMC data set and at 3 dpi in the WBC data set (Fig. 3B). Furthermore, in PBMC data set, a downregulation of the inflammatory BTM M37.0 was observed (Fig. 3A). BTMs related to the antigen presentation-related modules were mainly induced in D7/D3 (in PBMCs, listed in Fig. 3A) and D6/D3 (in WBCs, listed in Fig. 3B) comparisons. IC89 infection consistently activated the M165 module in PBMCs and WBCs (Fig. 3A and B, respectively), whereas the M40 and M139 modules were consistently activated in MA08 infection (Fig. 3A and B).

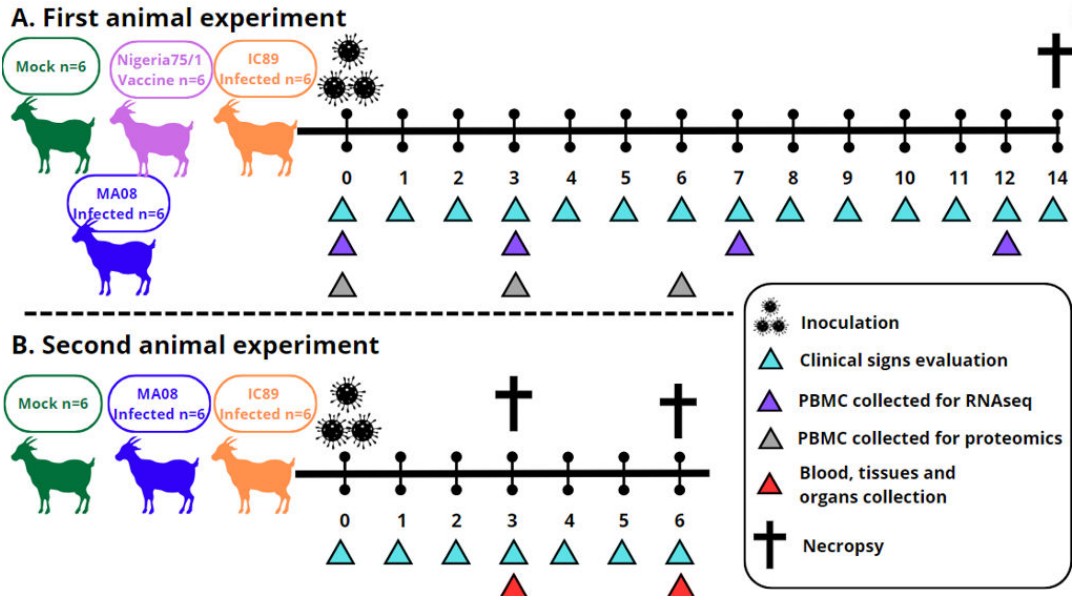

**FIG 1** Graphical summary of animal experiments included in this study.

## Tonsils

In the tonsils, two modules linked to the IFN-I response (M68 and M150, Fig. 3C) were activated early at 3 dpi, confirming the earlier induction of an IFN-I response in the MA08-infected animals. Despite this, the IFN-I response was comparable for the two PPRV strains. In contrast, only with the MA08 strain, a significant activation of many BTMs related to antigen presentation, inflammation, and myeloid cells was observed in both the D6/D0 and D6/D3 comparisons. For the inflammation, this included BTM related to inflammatory cytokines responses (M16, M33, M53, M37.0, and M24), chemoattraction of inflammatory cells (M132, M11.2, and M27.0), complement activation (M112.0), and coagulation (M11.1 and M85) (Fig. 3C).

## Mesenteric lymph node

In mesenteric lymph node, both infections mainly led to the induction of BTM related to type I IFN response (M13, M150, M68, and M111.0) and activation of the BTM M67 antigen-presenting module, associated with activated dendritic cells. These modules were activated at 3 or 6 dpi in animals infected with MA08 or IC89, respectively (Fig. 3D). Strikingly, neither inflammatory nor myeloid cell BTM response were found in this lymph node, suggesting that inflammation was restricted to the tonsillar mucosal barrier site at the time of sampling.

In conclusion, these results indicate that the IFN-I response is induced earlier by MA08 but does not appear to fundamentally differ from that induced by IC89. In contrast, major inflammatory, myeloid cell, and antigen-presenting cell responses were only induced in the tonsils by the MA08 strain.

## PPRV strain-dependent adaptive immunity transcriptional profiling

Adaptive immunity BTMs were divided into "cell cycle," "B-cell," and "T-cell" BTMs. The first and major response of B- and T-cell responses during an immune response is clonal expansion of antigen-specific lymphocytes, which is mediated by cell proliferation. This response is visible in the cell cycle BTM.

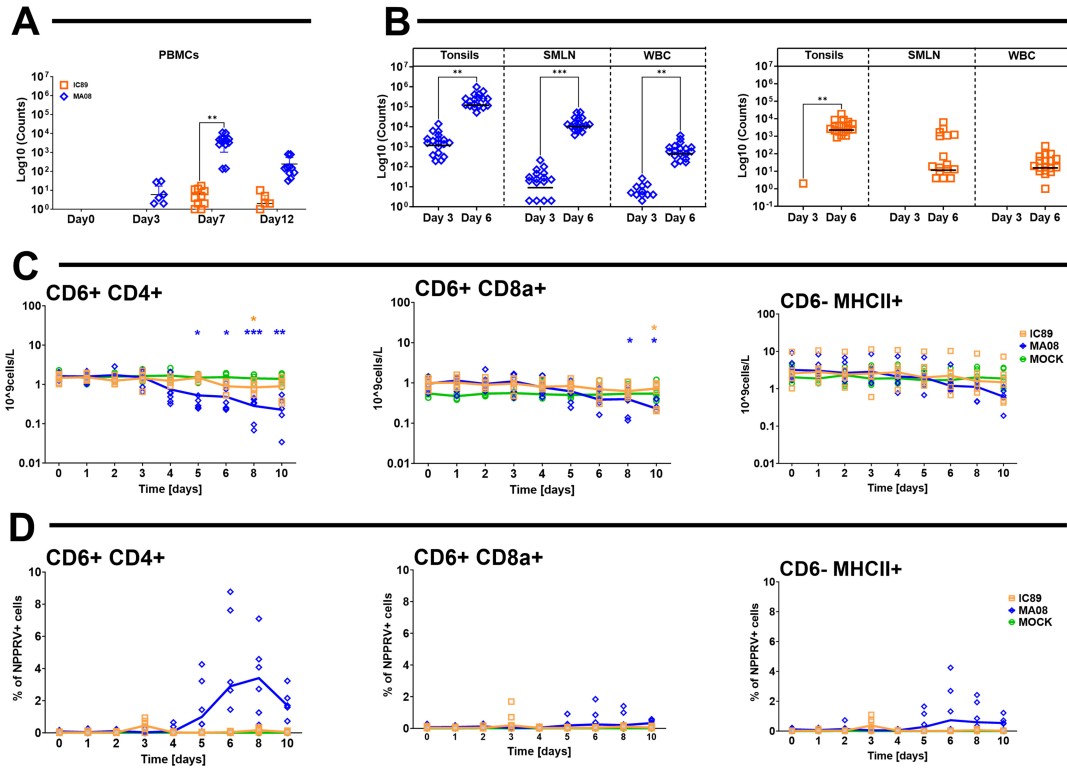

**FIG 2** Impact of infection and detection of viral genes and nucleoproteins In different organs and cell compartments. (A and B) The number of reads aligned to the viral genome in raw transcriptome data from PBMCs (A), tonsils, superior mesenteric lymph node (SMLN), and white blood cells (WBCs) (B) is shown. (C) The absolute numbers of CD4 T cells (CD4+CD6+), CD8a T cells (CD6+CD8a+), and antigen-presenting cells (CD6−MHCII+) in circulating PBMCs are shown. (D) Percentages of viral nucleoprotein in the PBMC populations are shown. In panel C, P values were calculated by comparing the group means of infected animals with those of mock-inoculated animals. *$P < 0.05$, **$P < 0.01$, ***$P < 0.001$.

## PBMCs and WBCs

In WBC and PBMC, cell cycle BTMs were initially downregulated at 3 and 6/7 dpi, followed by an upregulation at 12 dpi. While the downregulation of the cell cycle BTM was similar with the two PPRV strains, the upregulation at 12 dpi appeared more prominent with the MA08 strain (Fig. 4A and B). For B- and T-cell BTMs, we observed only a very few downregulated BTMs in the blood compartment.

## Tonsils and mesenteric lymph nodes

In the tonsils and the mesenteric lymph nodes, we observed a downregulation of cell cycle BTM at 6 dpi, which was much more prominent with the MA08 strain (Fig. 4C and D). In lymph nodes, we found no significant modulations with the IC89 strain (Fig. 4CD). With respect to B- and T-cell BTMs, similar to blood, very few B-cell BTMs were downregulated in the D6/D3 comparison in the lymph node following infection with MA08 (Fig. 4C and D).

In conclusion, the transcriptional profiles of adaptive immunity indicate a significant suppression of proliferation in the lymphoid tissue during the first week following infection with the MA08 strain only. This inhibition of proliferation is also observed in the blood cells and is only reverted at 12 dpi.

## Proteome analysis confirmed the induction of a stronger innate immune response during infection with the highly virulent strain

Proteomic data from PBMCs collected at 0, 3, and 6 dpi confirmed that both strains induced activation of intracellular sensors (RIG-I, OAS1, and TRIM25) and transcriptional

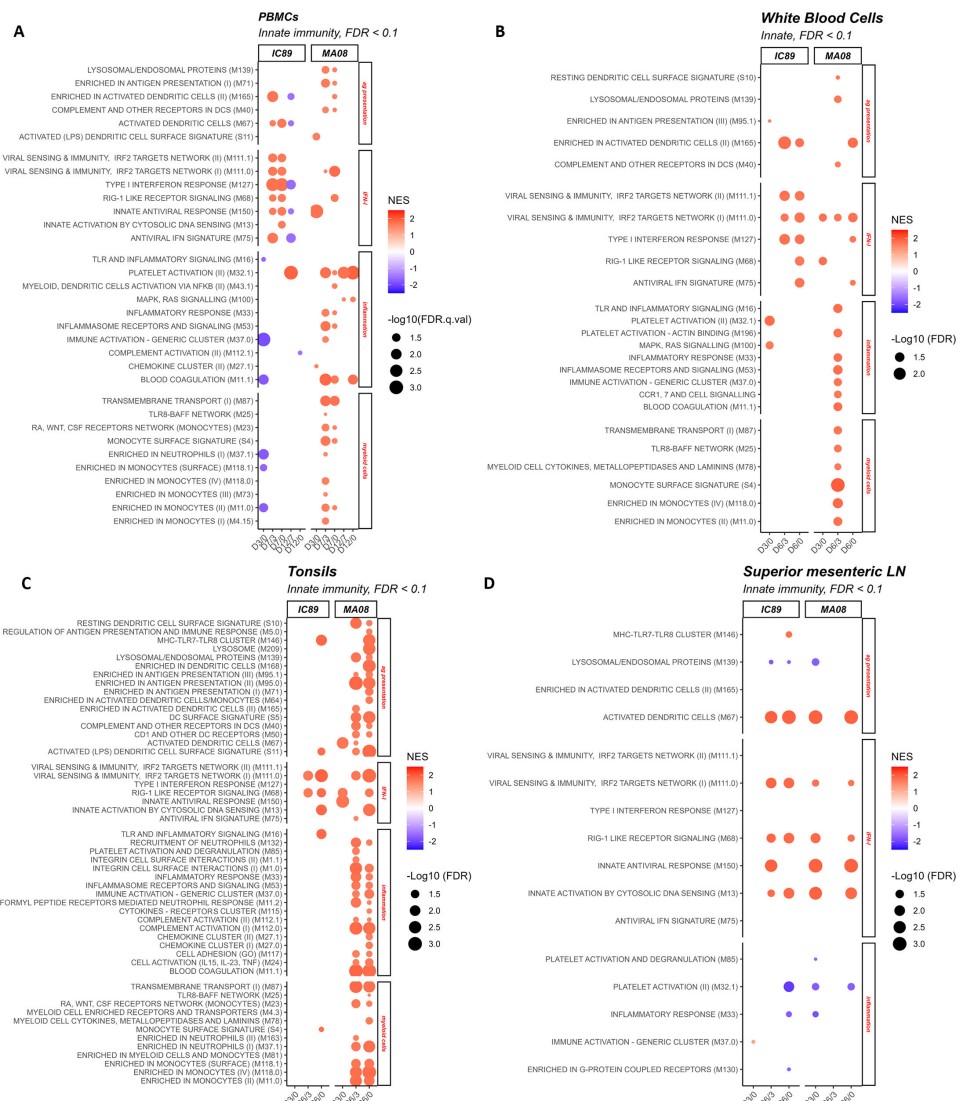

**FIG 3** Innate immunity-related BTM perturbations induced by IC89 and MA08 PPRV strains. These plots show the induction (red) or downregulation (blue) of innate immunity BTM expression in PBMCs (A), white blood cells (B), tonsils (C), and superior mesenteric lymph nodes (D) at 3 and 6 days post-infection. The intensity of the colors reflects the normalized enrichment scores (NES) determined after D0 to D3 (D3/D0), D3 to D7 (D7/D3), D0 to D7 (D7/D0), D12 to D7 (D12/D7), and D0 to D12 (D12/D0) comparisons for PBMCs, and D0 to D3 (D3/D0), D3 to D6 (D6/D3), and D0 to D6 (D6/D0) comparisons for other compartments. The size of the circles represents the false discovery rate (FDR) value. Families to which the significantly regulated BTMs correspond are shown on the right of the plots. D0 in these comparisons represents mock-inoculated animals sacrificed 3 days after infections.

factors such as EIF2AK2, although activation was consistently stronger during MA08 infection. Similarly, these recognitions of viral particles lead to earlier and more important synthesis of IFN-I-related proteins (CMPK2, DTX3L, STAT1, STAT2, MX1, MX2, IFIT1, IFIT2, IFIT3, ISG15, ISG20, NT5C3A, and WARS1) during MA08 infection.

In addition, MA08 infection increased the synthesis of ADAR, which enhances replication of RNA viruses in a direct or indirect manner, and of several inflammation-related proteins (ZBP1, LGALS3BP, S100A8, S100A9, S100A12, and HSPH1). However, the aminolevulinate dehydratase (ALAD) enzyme is only produced during IC89 infection. Unnamed proteins such as LOC102179380 and LOC102173185 appeared to be upregulated in both IC89 and MA08 infections (Fig. 5).

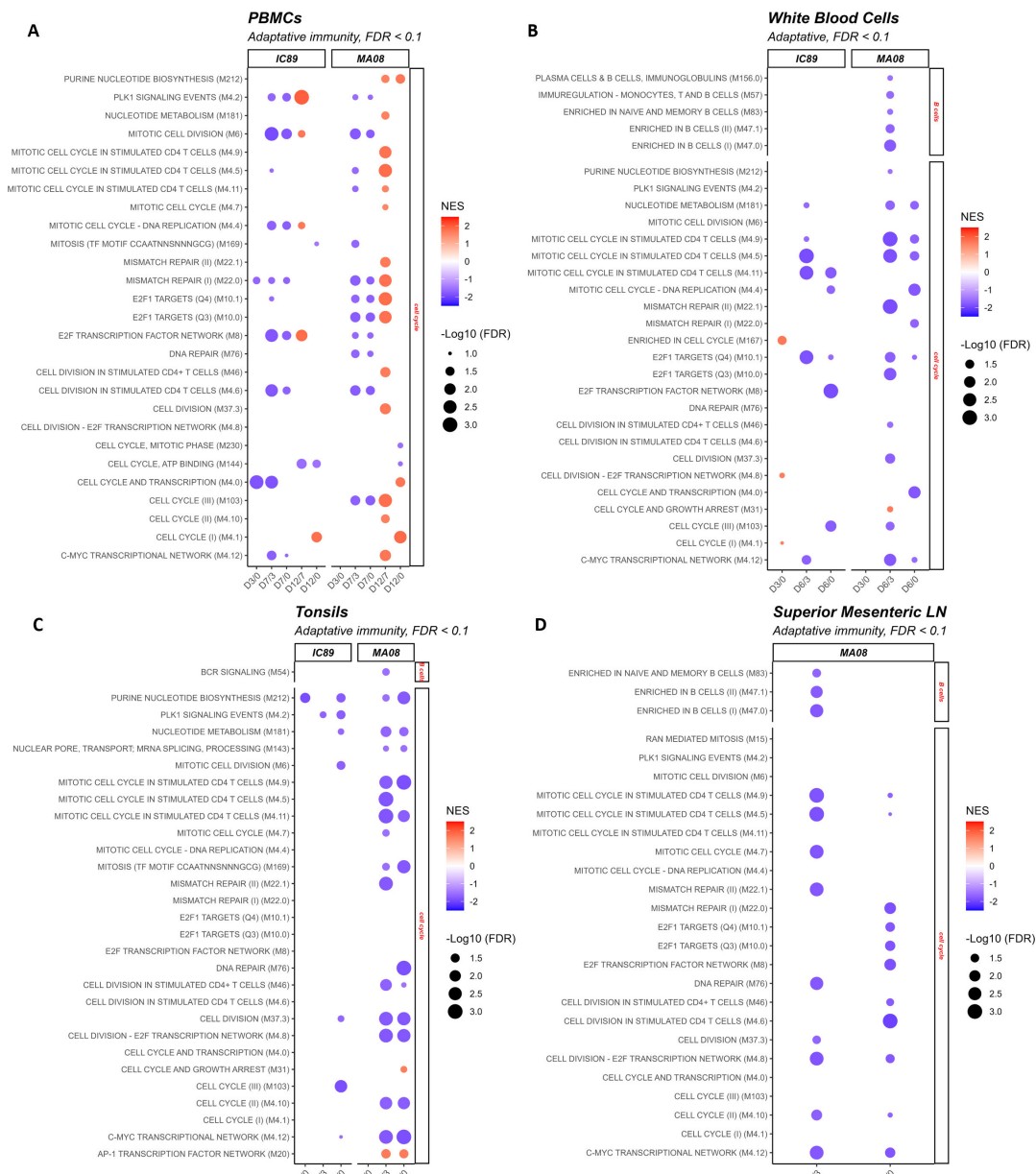

**FIG 4** Adaptative-immunity related BTM perturbations induced by IC89 and MA08 PPRV strains. These plots show the induction (red) or downregulation (blue) of adaptative immunity BTM expression in PBMCs (A), white blood cells (B), tonsils (C), and superior mesenteric lymph nodes (D) at 3 and 6 days post-infection. The intensity of the colors reflects the normalized enrichment scores (NES) determined after D0 to D3 (D3/D0), D3 to D7 (D7/D3), D0 to D7 (D7/D0), D12 to D7 (D12/D7), and D0 to D12 (D12/D0) comparisons for PBMCs, and D0 to D3 (D3/D0), D3 to D6 (D6/D3), and D0 to D6 (D6/D0) comparisons for other compartments. The size of the circles represents the false discovery rate (FDR) value. Families to which the significantly regulated BTMs correspond are shown on the right of the plots.

## PPRV infection of tonsillar epithelium by MA08 is associated with tonsillar inflammation

The transcriptomic analyses indicated severe inflammation and myeloid infiltration in the tonsils following MA08 infection. Considering that this effect was not observed in the MLN at the time of sampling, we postulated that PPRV infection in the tonsils could induce barrier defects that promote innate immune responses induced by bacteria present in the tonsillar crypts. Indeed, immunohistochemistry (IHC) for viral nucleoprotein demonstrated a high level of infection in the tonsillar epithelium in MA08-infected goats at 6 dpi, with a prominent positivity within the underlying lymphoid tissue.

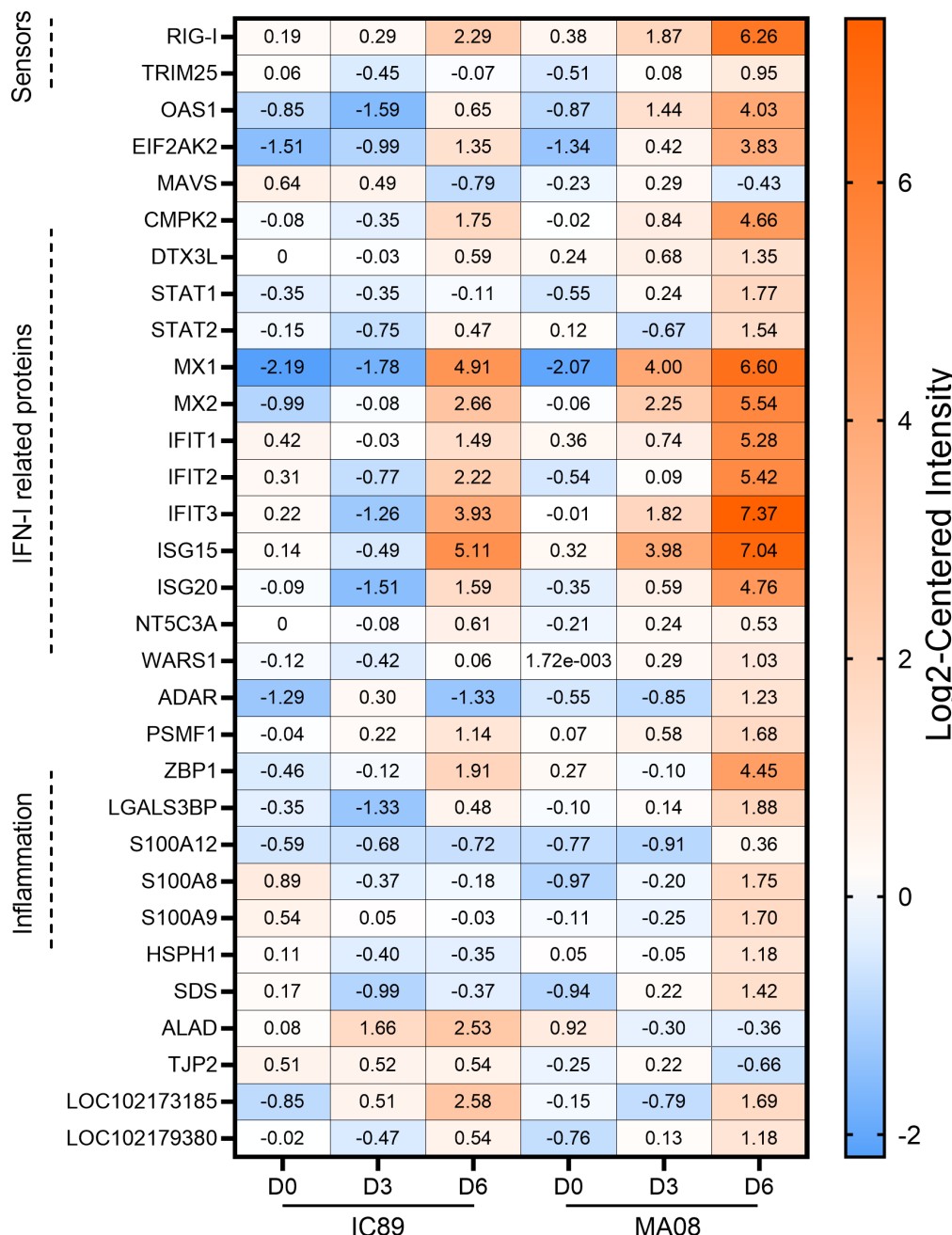

**FIG 5** Proteome regulation analysis during PPRV infection. Heatmap showing differentially expressed proteins in IC89 and MA08 infections at 0 (D0), 3 (D3), and 6 (D6) days post-infections. The intensity of the colors reflects the log2-centered intensity representing the abundance of a protein relative to the median value obtained for the same protein in all samples.

Moreover, the tonsillar epithelium showed histologically multifocal attenuation and ulcerations, which were multifocally associated with hemorrhages and fibrin exudation, as well as abundant leukocytes, predominantly neutrophils, admixed with bacteria in the lumen of the crypts (Fig. 6 and 7). Although aggregates of bacteria were also present in the lumen of the tonsils of control and IC89-infected animals, they showed no obvious epithelial damage or prominent pieces of evidence of acute inflammation. It is conceivable that the PPRV infection could cause the observed inflammatory responses directly or indirectly by disturbing the epithelial barrier function. In contrast, this inflammatory response was absent in MLN, despite high viral loads in MA08-inoculated goats at 6 dpi (Fig. 8).

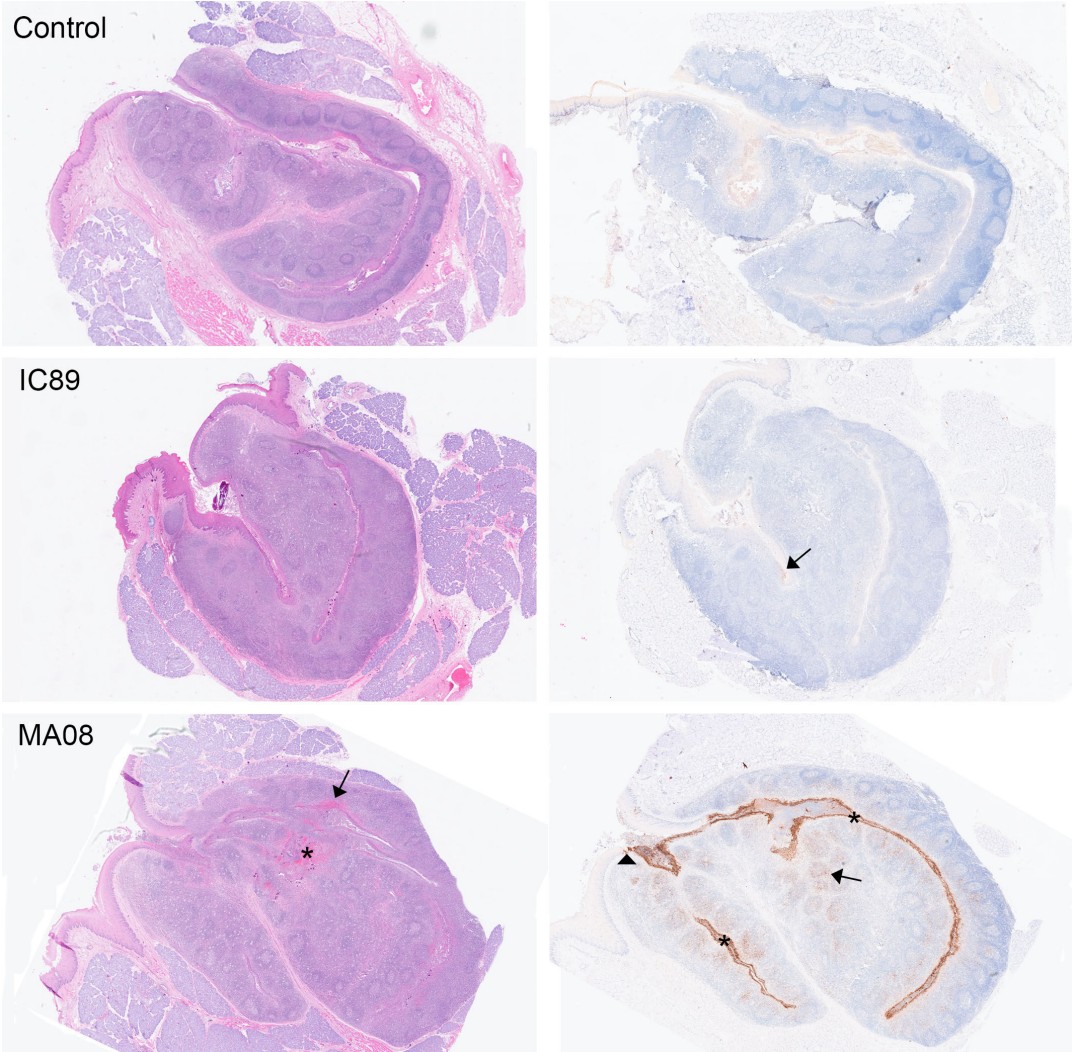

**FIG 6** Panel showing histological and immunohistochemistry (IHC) pictures of tonsils from one control, one IC89-inoculated and one MA08-inoculated goat at 6 dpi, representative of each experimental group. (Left) Hematoxylin and eosin. Multifocal hemorrhages are visible within the tonsillar lymphoid tissue (asterisk) and in damaged tonsillar epithelium (arrow) in the MA08-inoculated goat at 6 dpi. (Right) IHC for PPRV protein N. Very strong and diffuse intracytoplasmic staining (brown) is observed in the tonsillar epithelium of the MA08-infected goat (asterisks), with a sharp demarcation with the oropharyngeal epithelium (arrowhead). A prominent IHC PRRV positivity is also observed in the underlying follicular and parafollicular tonsillar tissue (arrow). In contrast, only weak multifocal positivity (arrow) is observed in the tonsillar epithelium and underlying lymphoid tissue in the IC89-infected goat, while the control is negative.

The IHC intensity and distribution within the lymphoid tissue in MA08-inoculated goats were similar in the tonsils and the MLN. Thus, at 6 dpi, there was a moderate to marked positivity in both follicular and parafollicular areas. In the tonsil, the strongest positivity was seen within the crypt epithelium of the tonsils. On the contrary, at 6 dpi, IC89-innoculated goats showed only few scattered stained individual cells within the lymphoid tissue, with few small, multifocal positively stained areas in the tonsillar crypt epithelium.

## DISCUSSION

To understand the biological basis of strain-dependent PPRV virulence, we have used an experimental model, based on the infection of Saanen goats or their PBMC with PPRV strains of different virulence (7, 26). In goats, the MA08 strain induced higher clinical scores associated with more prominent lymphocyte depletion and lesions particularly

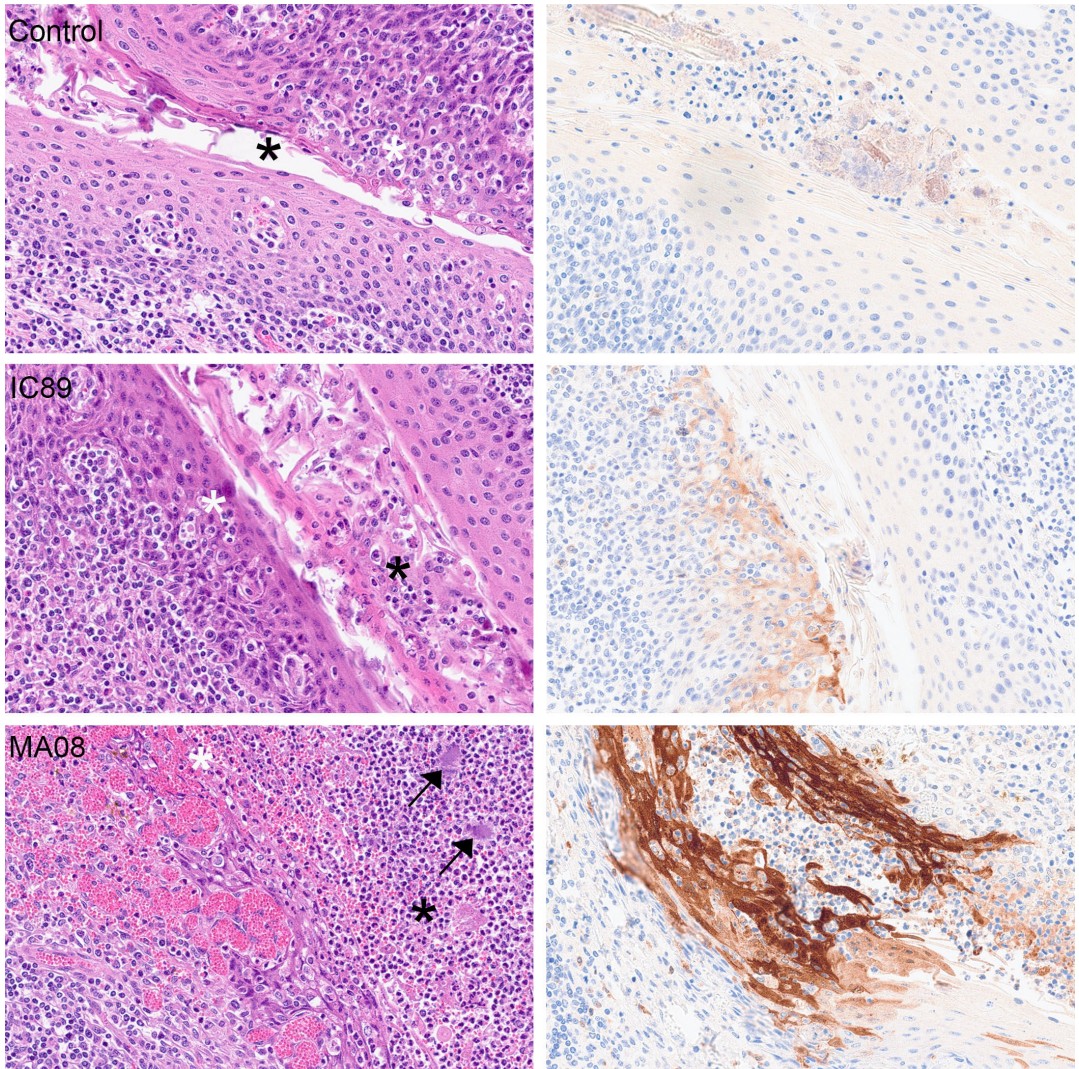

**FIG 7** Panel with higher magnification images showing details of the tonsillar epithelium from Fig. 6. (Left) Hematoxylin and eosin. Both control and IC89-infected goat show presence of mostly mononuclear leukocytes multifocally within the intact tonsillar epithelium (white asterisks) and low numbers of leukocytes in the lumen of the tonsillar crypts (black asterisks). MA08-infected goat shows multifocal attenuation and ulceration of the tonsillar epithelium associated with areas of hemorrhage (white asterisk), abundant neutrophils (black asterisk) and bacteria (arrows). (Right) IHC for PPRV protein N. The control is negative. A weak and strong positive brown staining is observed in the tonsillar epithelium of the IC89 and MA08-infected goats, respectively.

affecting mucosal and lymphoid tissues. As an *in vitro* model, we have used PBMCs from Saanen goats that were stimulated with mitogen and then infected with either IC89 or MA08. This revealed that viral replication prevents *in vitro* lymphocyte proliferation and promotes cell death (27). The present work aims to understand early events in viral immunopathogenesis related to these virulence-dependent features. To this end, we focussed on selected lymphoid tissues either in direct contact with mucosal surfaces (tonsils) or in lymph nodes draining mucosal tissue (MLN).

The nasopharyngeal epithelium and tonsillar tissues represent the likely initial entry site for PPRV, where the virus first replicates in epithelial cells followed by antigen-presenting cells (28, 29). Thereafter, infected antigen-presenting cells will reach lymph nodes draining the inoculation site (30). Our results are in line with this, showing that tonsils have the highest viral load in terms of viral RNA, starting at 3 dpi for the highly virulent MA08 and at 6 dpi for the low-virulent IC89 strain. Accordingly, following MA08 infection, innate BTMs, including inflammation, myeloid cells, and antigen presentation-related

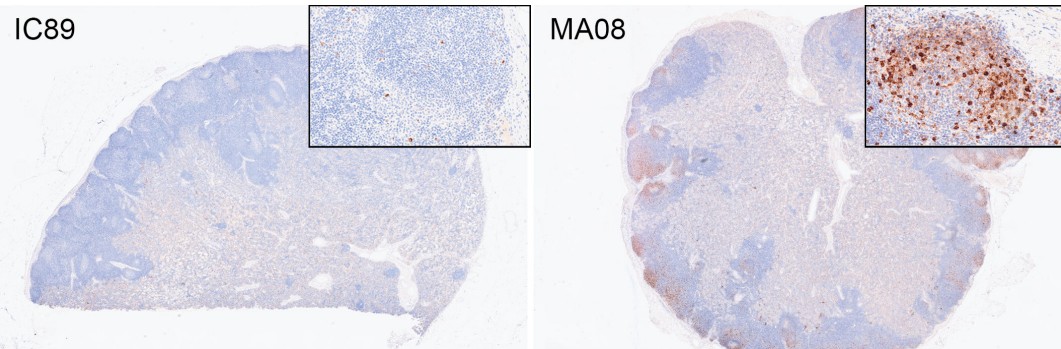

**FIG 8** Immunohistochemistry for PPRV in mesenteric lymph nodes from one IC89-inoculated and one MA08-inoculated goat at 6 dpi, representative of both experimental groups. IC89 showed only a mild scattered intracytoplasmic brown staining within the lymph node parenchyma, while MA08 showed abundant positivity, which seems to be more prominent but not restricted to the lymphoid follicles.

modules, are strongly activated in tonsils. However, it is quite remarkable that this inflammatory response was completely absent in MLN, despite high viral loads and a prominent presence of PPRV protein N at 6 dpi. A possible explanation is that a massive infection of the tonsillar epithelium by PPRV associated with the observed tonsillar epithelial damage leads to barrier defects, promoting inflammatory responses. Activation of inflammation-related modules may either be directly linked to the strong presence of PPRV protein N, which plays an important pro-inflammatory role by modulating inflammasome assembly (18). Alternatively, the detected inflammation may be induced by the bacteria, which are often being seen in the lumen of the crypts, being triggered by the breach in the tonsillar epithelium and therefore loss of this important anatomical and immunological barrier. A combination of both phenomena is also possible. In contrast, the MLN does not constitute a direct mucosal barrier. It can, however, not be totally ruled out that a similar inflammatory response and epithelial damage may occur in the intestine. Additional experiments would be needed to resolve this question. Animals infected with the less virulent strain showed neither damage in the tonsillar epithelium nor such massive infection or activation, which suggests that the anatomical and immunological barrier operates at the site of entry in the tonsillar crypts. It must be noted that in formalin-fixed tissues from natural measles cases, the majority of measles virus-infected cells in epithelium were the intraepithelial CD11c+ myeloid cells (31). Furthermore, it has been speculated that resident myeloid cells in the respiratory mucosa can transport PPRV (21). Future experiments are needed to understand the role of epithelial and myeloid cells in the inflammatory responses associated with highly virulent PPRV only. For example, differences in viral tropism for specific cell types might play a role in virulence.

Besides the induction of strong inflammatory responses in tonsils, our transcriptomic data also revealed an early activation of IFN-I responses at 3 dpi with the highly virulent MA08 strain. Nevertheless, at 6 dpi, the IFN-I related BTMs were expressed at levels comparable to those induced by IC89. This indicates that IFN-I response during PPRV may be not involved in pathogenic effects such as severe inflammation and lymphoid depletion.

Our results also demonstrate another prominent effect of virulent PPRV strains on the lymphoid tissue, which was the downregulation of the cell cycle BTM in the first week of infection. Proliferative responses in lymphoid tissue are of central importance because they indicate the initiation of adaptive immune responses and correlate well with clonal expansion and the induction of potent T-cell responses. Based on this, cell cycle BTMs have been demonstrated to represent a correlate of T-cell responses (32–35) and have been found to be induced in the first week of a virus infection (35, 36). Based on the opposite effect induced by PPRV and observations from other morbilliviruses (11–16), we propose that the suppression of these responses is part of the pathogenesis

of virulent PPRV infection. The cell cycle arrest may be linked to the significant reduction in circulating CD4[+] T lymphocytes observed in goats at 6 dpi with the MA08 strain. One possible pathway for inducing this immunosuppression may involve PPRV-infected dendritic cells, which have been shown to suppress the proliferative T-cell response (15). This may be in line with the observed depletion of CD4[+] T lymphocytes reported by others (37). In addition, we now also show that CD4[+] T cells are a main target of PPRV infection in the blood circulation during virulent PPR, and this could contribute to immunosuppression.

It is important to note that goats can overcome the immunosuppression induced by virulent PPRV. On one side, MA08 did not prevent the induction of neutralizing antibodies, which were first detected at around 7 dpi (7) . On the other hand, cell cycle BTMs were induced at 12 dpi. This suggest that neutralizing antibodies are essential and not sufficient in controlling this infection and the importance of the induction of PPRV-specific T-cell responses. Evidently, the general health of the animal at the time of infection will influence the efficacy of its immune response.

Our analysis of the proteome of PBMCs confirmed that infections are associated with the innate antiviral response in both strains, although more pronounced in the case of the highly virulent strain. It has also been shown that after vaccinating goats with the Sungri 96 PPRV vaccine, a majority of proteins associated with the antiviral interferon response were synthesized (38). The protein ADAR was differentially accumulated in both infections. This protein has a range of pro- or antiviral effects that can contribute to viral replication (39). Measles virus showed its ability to activate ADAR1, which affected the formation of stress granules and was beneficial for viral replication (40). Our results suggest that ADAR may participate in the differential replication of PPRV strains used in this study, without affecting the formation of stress granules (41). The ALAD enzyme has only been produced during IC89 infection. ALAD is known to regulate proteasome activity (42), and its activity is negatively correlated with oxidative stress in several pathologies such influenza B, where ALAD activity is lower in infected individuals compared to controls (43). Our results might suggest that ALAD activation could be a sign of controlled disease in IC89 infection, but this assertion needs to be investigated experimentally. The inflammation observed in MA08-infected animals may also be linked to protein expression, as a core set of inflammatory proteins was expressed in PBMCs. In this context, this strain is able to use all the mechanisms described elsewhere that are involved in PPRV virulence (17, 19, 44–48).

In conclusion, the present comparison of low and high virulent PPRV strains using a multiomics approach has identified crucial early innate and adaptive immunopathological processes associated with severe inflammatory disease and ineffective control of the virus by adaptive immune responses. This information is required to understand strain- and host-dependent differences in the pathogenesis of PPRV, which are relevant to PPRV surveillance and control strategies

## MATERIALS AND METHODS

### Animal experiments

Samples used in this study were collected during two different animal experiments. The first animal experimentation was previously described (7). Briefly, 24 Saanen goats aged between 2 and 12 years were randomly distributed into four groups (mock-inoculated, vaccinated, IC89-infected, and MA08-infected) of six animals. Infected animals were inoculated with IC89 or MA08 strains at day 0 (D0) with a total of 4 $\log_{10}$ TCID$_{50}$ per animal by the intranasal route (1 mL/nostril) using a specific intranasal mucosal atomization device (LMA MAD Nasal).

Clinical signs as well as the gross pathology and seroconversion of the animals were described. Blood samples were collected in all groups at days 0, 3, 7, and 12 post-infection to study the modulation of RNA over the time course of the experiment (Fig. 1A). Based on data obtained in this first experiment (see below), we decided to further

investigate transcriptional changes in different lymphoid organs at the early stage of infection with PPRV strains of different virulence. For this purpose, 18 adult Saanen goats were randomly distributed into three groups (mock inoculated, IC89 infected, and MA08 infected) of six animals. Macroscopical and histological PPRV-related lesions are detailed in Tables S1 and S2. Tonsils, superior MLN, and WBCs were collected at 3 and 6 days after inoculation. For each sampling day, three animals were sacrificed for organ collection. Clinical signs of disease were monitored throughout the experiment (Fig. 1B).

## Flow cytometry

PBMCs collected from the first experiment and prepared as previously described (27) were analyzed to quantify T-helper cells, defined as $CD4^+CD6^+$, cytolytic T cells, defined as $CD8^+CD6^+$, and antigen-presenting cells, defined as $CD6^-MHCII^+$. The latter contained mainly B cells, monocytes, and dendritic cells. Primary antibodies used were anti-CD6 (IgG1, clone BAQ91A), anti-CD4 (IgG2a, clone GC1A), anti-CD8α (IgG2a, clone 7C2B), and anti-MHCII (IgG2a, clone TH16A), all purchased from the Monoclonal Antibody Center (Washington State University, USA). Secondary antibodies conjugated with PE-Cy7 (anti IgG1) and Alexa Fluor 647 (anti IgG2a) were purchased from Thermo Fisher Scientific. For the intracellular staining of the virus, cells were fixed in 4% formalin solution paraformaldehyde (PFA) and permeabilized in 1× phosphate-buffered saline (PBS) supplemented with 0.1% of saponin. Fixed and permeabilized cells were stained with fluorescein isothiocyanate-conjugated nucleoprotein of peste des petits ruminants virus (49) antibodies (clone 38–4; CIRAD, Montpellier, France) diluted (1:100) in PBS supplemented with 0.3% of saponin.

## Histopathology and immunohistochemistry

Tonsil and MLN samples were placed in 10% buffered formalin, routinely processed for histology, sectioned at 3 µm, and stained with hematoxylin and eosin (H&E). For IHC, 3 µm formalin-fixed, paraffin-embedded tissue sections were mounted on positively charged slides (Color Frosted Plus; Biosystems, Muttenz, Switzerland), dried for 35 min at 60°C, and subsequently dewaxed, pretreated, and stained on Bond-III immunostainers (Leica Biosystems, Melbourne, Australia). After dewaxing (Bond Dewax solution, Leica Biosystems), the slides were subjected to a heat-induced epitope retrieval step using a Tris-EDTA-based buffer (Bond Epitope Retrieval 2, pH 9) for 10 min at 95°C to 100°C. To reduce non-specific binding of primary antibodies, a protein block solution was applied for 10 min at room temperature, as for all following steps. Then the slides were incubated with a primary mouse monoclonal, antibody against the PPRV nucleocapsid, clone 38–4 (49), at 1:100 for 15 min. All further steps were performed using reagents of the Bond Polymer Refine Detection Kit (Leica Biosystems) as follows: endogenous peroxidase was blocked for 5 min, then a rabbit-antimouse secondary antibody was applied (8 min), followed by a peroxidase-labeled polymer (8 min). Both of these reagents were supplemented with 2% dog serum to block non-specific binding (LabForce, Nunningen, Switzerland). Finally, slides were developed in 3,3′-diaminobenzidine/$H_2O_2$ (10 min), counterstained with hematoxylin, and mounted. In negative controls, the primary antibody was replaced with wash buffer. Known positive controls were stained in parallel with each series.

## RNA sequencing

Total RNA was extracted from caprine PBMCs, WBCs, and organs using the protocol described in Eloiflin et al. (27). Total RNA quality and concentration were measured, and the library was prepared at the Next Generation Sequencing platform of the University of Bern, Switzerland. Only samples with an RNA quality number score of ≥8 were sequenced on a NovaSeq 6000 instrument using an SP flow cell. Raw sequencing reads were aligned on either *Capra hircus* ARS1 genome (GCA_001704415.1) or PPRV genome (GCA_000866445.1 ViralProj15499) using STAR 2.7.3a (50). Before the mapping

step, STAR index was generated from FASTA and annotation GTF files of selected references with the following parameters: star --runThreadN –runMode --genomeDir --genomeFastaFiles --sjdbGTFfile --sjdbOverhang. Read mapping was then performed on the generated index using the following parameters: STAR—genomeDir –runThreadN –readFilesCommand --readFilesIn --outFileNamePrefix --outSAMtype BAM—outSAMun-mapped --peOverlapNbasesMin --outSAMattributes. Quantification of gene expression was performed using FeatureCounts v.2.0.1 (51). Reads were assigned to the reference annotations at the exon level, and counts were summarized at the gene level using default parameters (-t exon -g gene_id), corresponding to unambiguously assigning uniquely aligned paired-end (-p) and multimapped reads (-M). Then, the obtained count table was used for differential expression analysis between non-infected (mock-inocu-lated) and infected animals at 3 and 6 days post-infection with DESeq2 v.1.36.0 (52) on R software v.4.2.2. Before running the differential expression analysis, clustering of replicates was verified by running a principal component analysis with the plotPCA function of the DESeq2 package.

## Gene set enrichment analysis

Differentially expressed gene lists obtained after comparisons were ranked to enable gene set enrichment analysis (GSEA) using GSEA v.4.3.0 desktop application (53, 54). This ranking assigns each gene a unique rank value based on its expression value (log2 fold changes) and its adjusted $P$ value. Thus, each gene's rank is calculated using the following formula: $RNK = -\log$ (adjusted $P$ value) $\times$ SIGN, where SIGN is "−1" when the gene is downregulated and "+1" when it is upregulated. These ranked files were then loaded into the software. BTMs adapted to *Capra hircus* as well as a chip annotation file that lists each identifier and its matching HGNC gene symbol used in BTM (27) were also loaded in the software and used first to run the Chip2Chip mapping tools with default parameters. This step converts gene sets to the format required for the chip platform and allows analysis of the data set without reducing the probe sets to gene symbols. Ranked files and BTMs mapped to the chip platform were used to run the GSEAPreranked mode with the following parameters: required fields: -- Number of permutations 1000; --Collapse/Remap to gene symbols No_Collapse and basic fields: --Enrichment statistic weighted; --Max size: exclude larger sets 500; --Min size: exclude smaller sets 5. All other fields were used with the default parameters. BTMs were classified in families relating to "antigen presentation," "IFN type I," "myeloid cells," "B cells," "T cells," and "cell cycle" as previously described (27). The ES and adjusted $P$ values for the BTM were visualized with ggplot2 in R software.

## Proteomics

During the first animal experiment and after extraction, PBMCs were also collected in DIGE buffer (7 M urea, 2 M thiourea, 4% 3-[(3-cholamidopropyl)-dimethylammonio]-1-propanesulfonate and 30 mM Tris-HCL; pH 8). These samples were prepared and analyzed by mass spectrometry following the protocol described in Eloiflin et al. (27). After the mass spectrometry analysis, MaxQuant's label-free quantification (LFQ) data were log2 transformed and filtered, and missing data were imputed according to the normal distribution (width = 0.3 and offset = 1.8). Log2 LFQ data were used to iden-tify differentially expressed proteins in our samples by calculating their log2-centered intensity values using the following formula:

Log2-centered intensity$_i$ = log2 LFQ$_i$– median ({log2 LFQ$_1$, log2 LFQ$_2$, …}),

where $i$ represents the protein of interest in a specific condition and {log2 LFQ$_1$, log2 LFQ$_2$, …} represents the set of log2 LFQ values obtained for the protein of interest under all the conditions tested. A heatmap of the log2-centered intensity values for the most significantly expressed proteins between conditions was represented using GraphPad Prism v.9.4.1.

## Statistical analysis

Statistical tests were carried out with GraphPad Prism v.9.4.1 (GraphPad Software, California, USA). A mixed effects model (REML) was used with the Geisser-Greenhouse correction. In this model, the measures recorded over time for each of the animals were considered as matched values. $P$ values between the mean of groups (mock-, IC89- or MA08-infected animals) were calculated using Tukey's multiple comparison test. Individual's variances were computed for each comparison. Asterisks on the graphs highlight statistical differences between the comparisons: $*P < 0.05$, $**P < 0.01$, $***P < 0.001$, $****P < 0.0001$.

## ACKNOWLEDGMENTS

The authors thank the staffs of the Functional Proteomics Platform (Biocampus Montpellier), UMR ASTRE (CIRAD) and the Institute of Virology and Immunology (BERN) for their help with this project.

Conceptualization: R.-J.E., P.T., O.G.-N., A.S., and A.B.; data curation: R.-J.E. and A.S.; formal analysis: R.-J.E., L.G.-R., and A.S.; funding acquisition and supervision: A.B. and A.S.; investigation: R.-J.E., V.L., S.P., O.G.-N., and L.G.-R.; methodology: R.-J.E., S.T., O.G.-N., L.G.-R., P.T., and A.S.; validation: P.T. and A.S.; visualization: R.-J.E. and L.G.-R.; writing (original draft preparation): R.-J.E.; writing (review and editing): R.-J.E., L.G.-R., V.L., S.P., S.T., P.T., O.G.-N., A.S., and A.B.

## AUTHOR AFFILIATIONS

[1]ASTRE, University of Montpellier, CIRAD, INRAE, Montpellier, France
[2]CIRAD, UMR ASTRE, Montpellier, France
[3]INTERTRYP, Univ Montpellier, IRD, CIRAD, Montpellier, France
[4]Institute of Animal Pathology, COMPATH, Department of Infectious Diseases and Pathobiology, Vetsuisse Faculty, University of Bern, Bern, Switzerland
[5]Institute of Virology and Immunology, Mittelhäusern, Switzerland
[6]Department of Infectious Diseases and Pathobiology, Vetsuisse Faculty, University of Bern, Bern, Switzerland

## AUTHOR ORCIDs

Roger-Junior Eloiflin  http://orcid.org/0000-0001-5927-5592
Arnaud Bataille  http://orcid.org/0000-0002-3508-2144

## FUNDING

| Funder | Grant(s) | Author(s) |
|---|---|---|
| European Commission (EC) | 731014 | Artur Summerfield |
| | | Arnaud Bataille |

## AUTHOR CONTRIBUTIONS

Roger-Junior Eloiflin, Conceptualization, Data curation, Formal analysis, Investigation, Methodology, Project administration, Validation, Visualization, Writing – original draft, Writing – review and editing | Llorenç Grau-Roma, Formal analysis, Investigation, Methodology, Visualization, Writing – review and editing | Vincent Lasserre, Investigation, Writing – review and editing | Sylvie Python, Investigation, Writing – review and editing | Stephanie Talker, Methodology, Writing – review and editing | Philippe Totte, Conceptualization, Methodology, Validation, Writing – review and editing | Obdulio García- Nicolás, Conceptualization, Investigation, Methodology, Writing – review and editing | Artur Summerfield, Conceptualization, Data curation, Formal analysis, Funding acquisition,

Methodology, Supervision, Validation, Writing – review and editing | Arnaud Bataille, Conceptualization, Funding acquisition, Methodology, Writing – review and editing

## DATA AVAILABILITY

Sequencing data have been deposited in National Center for Biotechnology Information Sequence Read Archive (PRJNA1103751).

## ETHICS STATEMENT

Animal experimentations were carried out in the high containment facility of the Institute of Virology and Immunology (Mittelhäusern), in accordance with the Swiss animal protection law (TSchG SR 455, TSchV SR 455.1, and TVV SR 455.163). The committee on animal experiments of the canton of Bern, Switzerland, reviewed the experiments, and the cantonal veterinary authority approved the study under the authorization number BE16/2020.

## ADDITIONAL FILES

The following material is available online.

### Supplemental Material

**Figure S1; Tables S1 and S2 (Spectrum03124-24-s0001.pdf).** RT-qPCR for virus detection in organs and tables summarizing erosive-ulcerous and histological lesions.

### Open Peer Review

**PEER REVIEW HISTORY (review-history.pdf).** An accounting of the reviewer comments and feedback.

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
