## [Reviewer comments · Microbiology Spectrum]

Microbiology Spectrum

Peste des Petits Ruminants virus virulence is associated with an early inflammatory profile in the tonsils and cell cycle arrest in lymphoid tissue

Roger-Junior Eloiflin, Llorenç Grau-Roma, Vincent Lasserre, Sylvie Python, Stephanie Talker, Phillippe Totte, Obdulio García-Nicolás, Artur Summerfield, and Arnaud Bataille

Corresponding Author(s): Roger-Junior Eloiflin, CIRAD Departement Systemes biologiques

Review Timeline:

Submission Date:	December 2, 2024
Editorial Decision:	December 13, 2024
Revision Received:	December 19, 2024
Accepted:	December 23, 2024

Editor: Artem Rogovsky

Reviewer(s): The reviewers have opted to remain anonymous.

Transaction Report:

DOI: <https://doi.org/10.1128/spectrum.03124-24>

Re: Spectrum03124-24 (Peste des Petits Ruminants virus virulence is associated with an early inflammatory profile in the tonsils and cell cycle arrest in lymphoid tissue)

Dear Dr. Roger-Junior Eloiflin:

Thank you for the privilege of reviewing your work. Below you will find my comments, instructions from the Spectrum editorial office, and the reviewer comments.

I am pleased to inform you that your manuscript has been editorially accepted for publication. However, there are a few additional questions in the submission form that need to be answered before the final decision. Once these are completed, please return your submission so that I can move your paper forward to acceptance.

Revision Guidelines

Sincerely,
Artem Rogovsky
Editor
Microbiology Spectrum

Re: Spectrum03124-24R1 (Peste des Petits Ruminants virus virulence is associated with an early inflammatory profile in the tonsils and cell cycle arrest in lymphoid tissue)

Dear Dr. Roger-Junior Eloiflin:

Your manuscript has been accepted, and I am forwarding it to the ASM production staff for publication. Your paper will first be checked to make sure all elements meet the technical requirements. ASM staff will contact you if anything needs to be revised before copyediting and production can begin. Otherwise, you will be notified when your proofs are ready to be viewed.

Sincerely,
Artem Rogovsky
Editor
Microbiology Spectrum